# 3D Geophysical Modeling Based on Multi-Scale Edge Detection, Magnetic Susceptibility Inversion, and Magnetization Vector Inversion in Panjshir, Afghanistan to Detect Probabilistic Fe-Polymetallic Bearing Zone

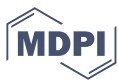

**Mohammad Hakim Rezayee [1,*], Ahamd Qasim Akbar [2], Torabaz Poyesh [3], Ezatullah Rawnaq [1], Khair Mohammad Samim [1] and Hideki Mizunaga [1]**

[1] Department of Earth Resources Engineering, Graduate School of Engineering, Kyushu University, 744 Motooka, Nishi Ward, Fukuoka 8190053, Japan; rawnaq.ezatullah.104@s.kyushu-u.ac.jp (E.R.); samim.khair.mohammad.162@s.kyushu-u.ac.jp (K.M.S.); mizunaga@mine.kyushu-u.ac.jp (H.M.)

[2] Department of Civil Engineering, Graduate School of Engineering, Kyushu University, 744 Motooka, Nishi Ward, Fukuoka 8190053, Japan; qasim1368@live.com

[3] Department of Soil Science and Irrigation, Agriculture Faculty, Bamyan University, Bamyan 1601, Afghanistan; poyesh04@gmail.com

\* Correspondence: rezayee.mohammadhakim.624@s.kyushu-u.ac.jp

**Abstract:** The Panjshir Fe-Polymetallic ore deposit is a valuable geological resource in Afghanistan, rich in iron and multiple essential metallic minerals, with substantial potential for industrial development. The exploration phase faces challenges related to the complex geological settings, high variability of mineral compositions, and the need for advanced geophysical techniques to accurately locate and assess valuable metallic resources. Considering the strong magnetic characteristics exhibited by Fe-Polymetallic elements, geomagnetic data were employed to analyze and map the likely prospectivity of Fe-Polymetallic deposits within the study area. Multi-scale edge detection techniques were employed to accurately map the boundaries of magnetic bodies by utilizing the upward continued analytical signal amplitude. The presence of a fault system on the geological map confirmed the structural information derived from our edge detection techniques. Advanced magnetic data inversion techniques were employed to create a three-dimensional representation of the distribution of magnetic bodies linked to Fe-Polymetallic deposits. In our efforts to reduce the impact of remnant magnetization in the study area, we adopted a comprehensive strategy by employing both magnetic susceptibility and magnetization vector inversion techniques. The use of a sparse and blocky norm regularization [0,1,1,1] is well-suited for magnetic susceptibility inversion, while a blocky norm [0000,0000,0000] is the appropriate choice for magnetization vector inversion in our study. Ultimately, the zones characterized by a high magnetic susceptibility and a high magnetization amplitude are considered promising areas for potential Fe-polymetallic occurrences.

**Keywords:** geophysical modeling; magnetic susceptibility; magnetization vector; inversion; Fe-polymetallic

## 1. Introduction

Polymetallic minerals refer to geological deposits that contain multiple valuable metals in varying proportions within the same ore body. For a variety of elements, such as cobalt, nickel, copper, titanium, and rare earth metals, polymetallic deposits have been recognized as extremely important potential economic resources [1–3]. These deposits are economically important to the mining industry because they are the primary sources of several metals. Currently, the most widely used metals globally are iron, copper, and zinc, ranked as the top three and fourth, respectively. The automotive, aerospace, construction, equipment, shipping, and other sectors depend on these metals as essential raw materials [4–6]. Metals

like lithium, cobalt, and rare earth elements, which are frequently found in polymetallic deposits are essential to modern electronics, electric cars, and renewable energy systems. The need for sustainable and safe metal resources is driven by the demand for these technologies. The need to find and sustainably use polymetallic resources is expanding as more and more sectors seek to meet their demands.

The majority of hydrothermally characterized polymetallic deposits are located close to magma-active areas. There are numerous faults, metal sulfides, and hydrothermal changes in these areas. During exploration, the prospecting depth is progressively raised to hundreds of meters [7]. Polymetallic ore deposits are efficiently mapped using various surveying techniques, allowing geologists and mining companies to comprehend the properties and potential of these priceless resources. The study presented in [8] models the polymetallic mineralization possibility using multi-geospatial data and logistic regression. To find anomalies in the concentration of metals, geochemical surveys involving collecting and analyzing samples of rocks are employed by [9] to prospect the Fe-Polymetallic mineralization. The work presented in [10] proposes the use of radon as a special tracer for identifying Fe-Mn nodule resources in the enormous abyssal regions of the world ocean by using it to map regional deep ocean ferromanganese nodule fields. In-depth geophysical studies are imperative to profoundly comprehend the intricate subsurface composition and configuration within the Earth's crust [11–13]. These extensive investigations play a fundamental role in enhancing our knowledge of the deep-seated structural aspects of the Earth's crust. Despite the facts mentioned above, metal sulfide concentrations were found, and information on below-ground media was gathered using geophysical exploration methods [14–17].

Magnetic surveying is a fundamental technique in mineral exploration, particularly when targeting Fe-Polymetallic units comprising a mix of iron and valuable metals like copper, zinc, lead, and more. This method relies on the Earth's natural magnetic field and variations caused by magnetic minerals beneath the surface. Numerous survey applications have used three-dimensional magnetic modeling, crucial for learning more about magnetic susceptibility distribution below the surface [18]. The field of magnetic inversion applications includes, but is not limited to, planetary geophysics [19,20], geological and tectonic studies [21,22], and mineral prospecting [23–26].

This lack of prior research in our assigned study area is mainly due to the small scale of deep exploration operations. This leads to a lack of understanding regarding the potential for polymetallic formation, as well as the underlying ore-controlling structures in this area. The Fe-Polymetallic deposit potential in our study region was evaluated by the application of magnetic surveying techniques. Multi-scale edge detection, magnetic susceptibility inversion, and magnetization vector inversion were among the techniques we used in our magnetic data analysis techniques. Our study begins with an examination of the geological background of the study area, followed by the sequential steps of magnetic data processing and inverse modeling. Subsequently, our findings are deliberated in the context of surface geology information, and our research is wrapped up in the concluding phase.

## 2. Geological Background

Our present understanding of the geology of the Panjshir Valley and Western Hindu Kush area is based on intensive research and geologic mapping conducted by Soviet and Afghan geologists throughout the 1960s and 1970s. Units that define the spatial range of various rock types were commonly indicated on a 1:250,000-scale and a smaller-scale map (Figure 1). The Middle Afghanistan Geosuture, as identified by the work presented in [27], extends westward through Bamyan, acting as a geological demarcation. It separates the platform sedimentary rocks in northern Afghanistan from the diverse southern structural blocks, playing a crucial role in delineating the geology of the Panjshir Valley and Western Hindu Kush region. This important crustal feature in the Panjshir Valley region is dominated by the Hindu Kush Tectonic Zone, a network of integrating faults that traverse northeast. A significant structural split between Precambrian rocks

of the East Afghanistan median massif and sedimentary rocks in the Panjshir Valley is represented by the Panjshir fault, which runs parallel to and a few miles southeast of the Hindu Kush Tectonic Zone [28,29]. The ages given to metamorphic and plutonic rocks across the area are subject to significant ambiguity [28,30]. Large portions of the Hindu Kush are considered Proterozoic or Archean in age based on their metamorphic grade, although some of these portions may be Devonian to cretaceous based on fossil data [31]. Amphibolite and highly foliated, layered gneisses with varying orientations are seen in the southwestern part of the valley, where the rock composition ranges from felsic to mafic. In fault contact with metasedimentary rocks to the northwest are this band of gneisses, which extends beyond the Panjshir Valley to the northeast. Exposed metasedimentary units are present between the two belts of gneisses and are exposed throughout. A second band of gneisses extends west of and parallel to the valley.

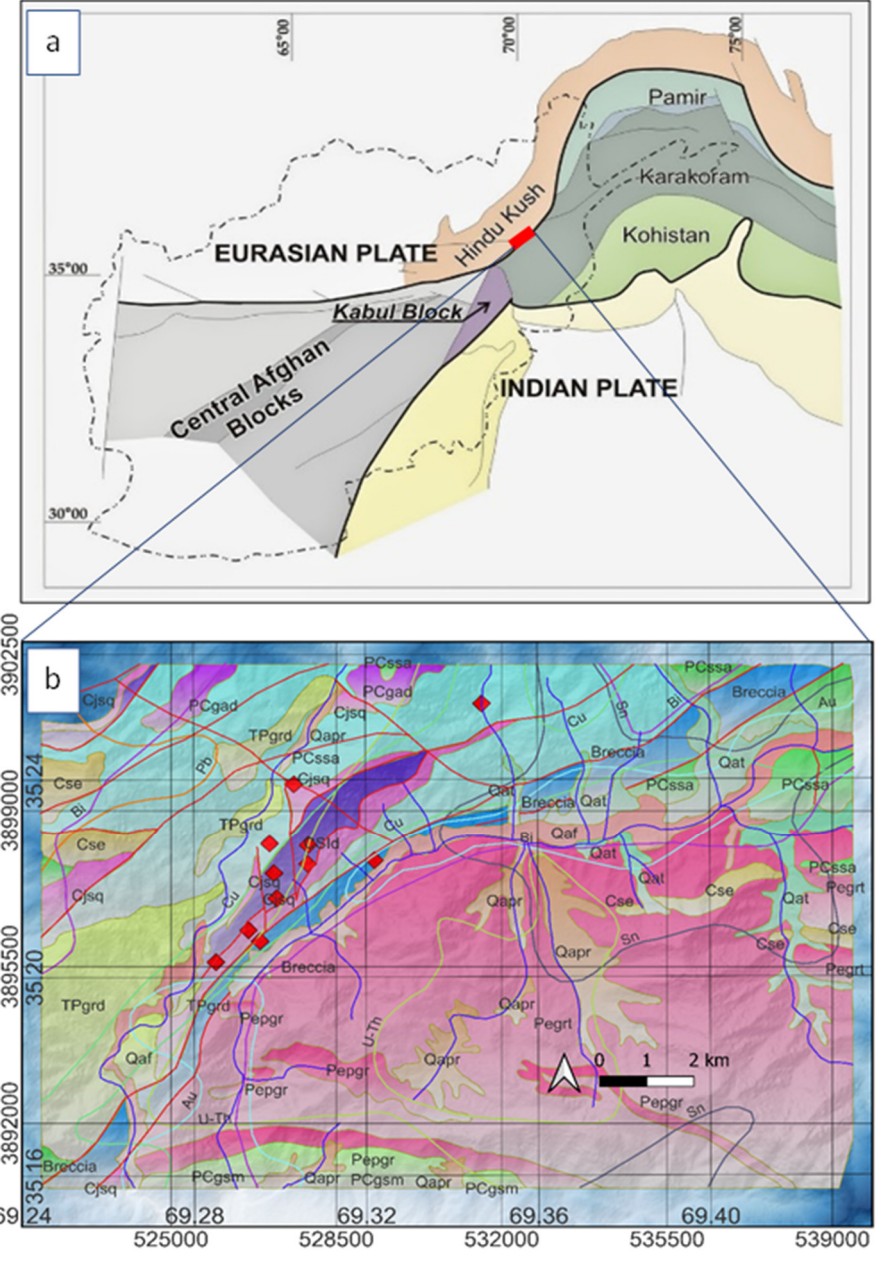

**Figure 1.** *Cont.*

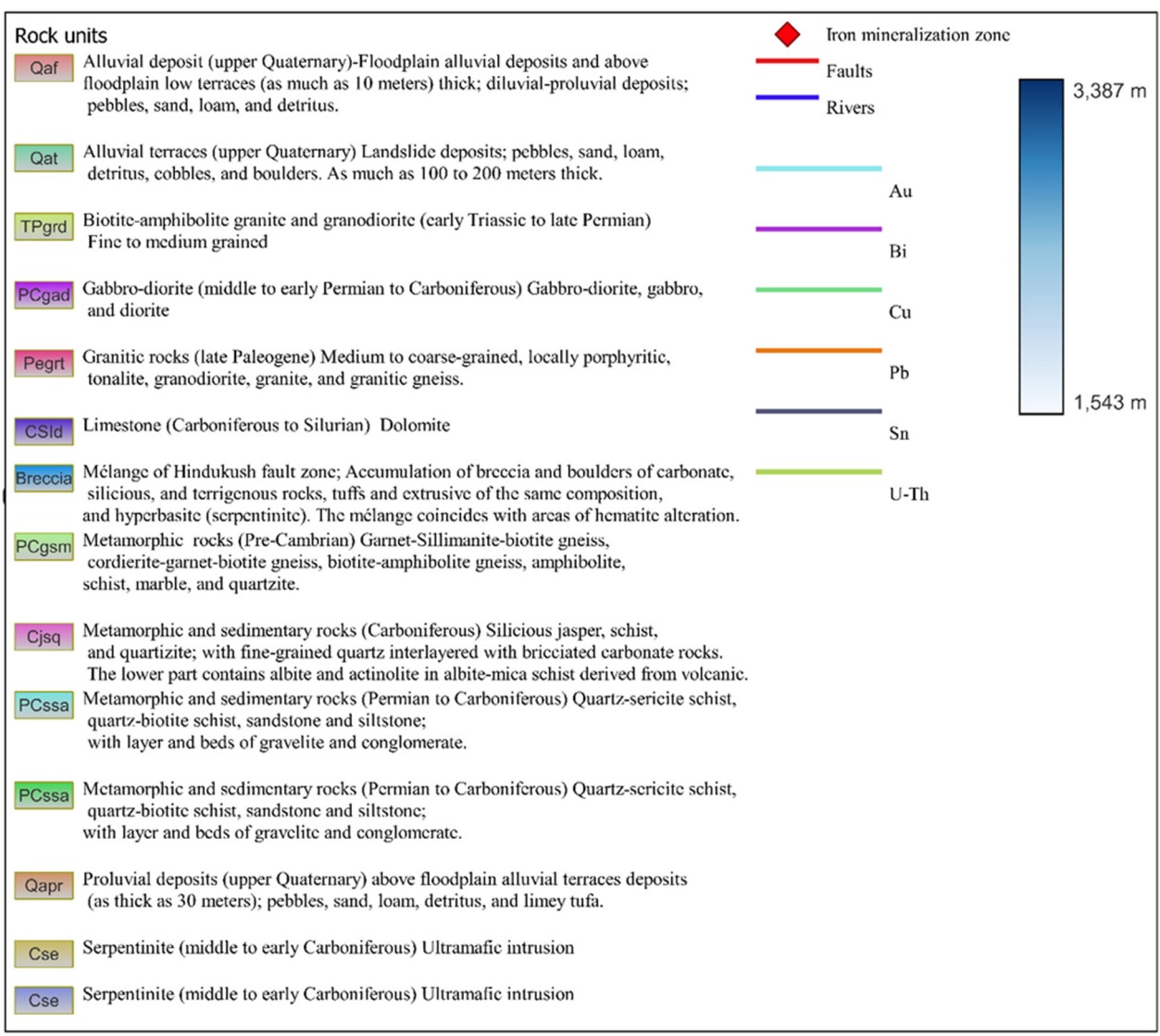

**Figure 1.** (**a**) The tectonic system of Afghanistan, and (**b**) The geological map is a redrafted and modified version of the Geological map and map of the mineral resources of the basin Ghorband, Salang, and Panjshir [32]. Here, only the Panjshir area is projected.

Figure 1 presents the spatial distribution of geological formations, encompassing various rock types across the area of interest. This illustration likely visually represents the geological diversity and distribution within the geological map. In the western central region of the map, specific zones with a higher concentration of iron are highlighted. These zones are associated with rock formations characterized by metamorphic and sedimentary rocks from the carboniferous period. These formations include various rock types, such as siliceous jasper, schist, and quartzite. Within these rock layers, there are occurrences of fine-grained quartz interlayered with brecciated carbonate rocks. Various geological materials are accumulated in the southeastern region of these iron-bearing zones. That includes the presence of breccia and large boulders composed of carbonate, siliceous, and terrigenous rocks. Additionally, there are deposits of tuffs and extrusive formations with similar compositions. Along the Hindukush fault zone within this area, hyperbasite rock types, particularly serpentinite, are prevalent.

There are no prior public reports or studies available concerning local geological mapping in the vicinity of our study area. To generate a comprehensive local geology map for the study area, we have placed specific emphasis on highlighting the iron-bearing zone. This emphasis has been achieved through the application of remote sensing techniques,

with a particular focus on satellite image processing. This approach involves analyzing and interpreting satellite imagery to gain valuable insights into the characteristics and distribution of iron-rich areas within this zone. A cloud-free level 1T ASTER image was downloaded from the U.S. Geological Survey Earth Resources Observation and Science Center (USGS EROS) "https://earthexplorer.usgs.gov (accessed on 20 October 2023)". A band combination of 4, 6, 8 is used to visualize the geological formation, and the sediment boundaries are divided based on a color variation of the sediments. Using remote sensing techniques, we have identified and delineated lithological boundaries within the area of interest [33–35]. Subsequently, we have refined and constrained these boundaries by utilizing a hyperspectral map from the work presented in [36], allowing us to gain a more detailed understanding of the mineral composition in the study area (Figure 2). The magnetic surveying activities that have been conducted are denoted by the black-colored box on the map. The predominant mineral composition in this area is primarily linked to calcite and dolomite-based sediments. Additionally, there are occurrences of goethite and iron minerals represented by both $Fe^{2+}$ and $Fe^{3+}$.

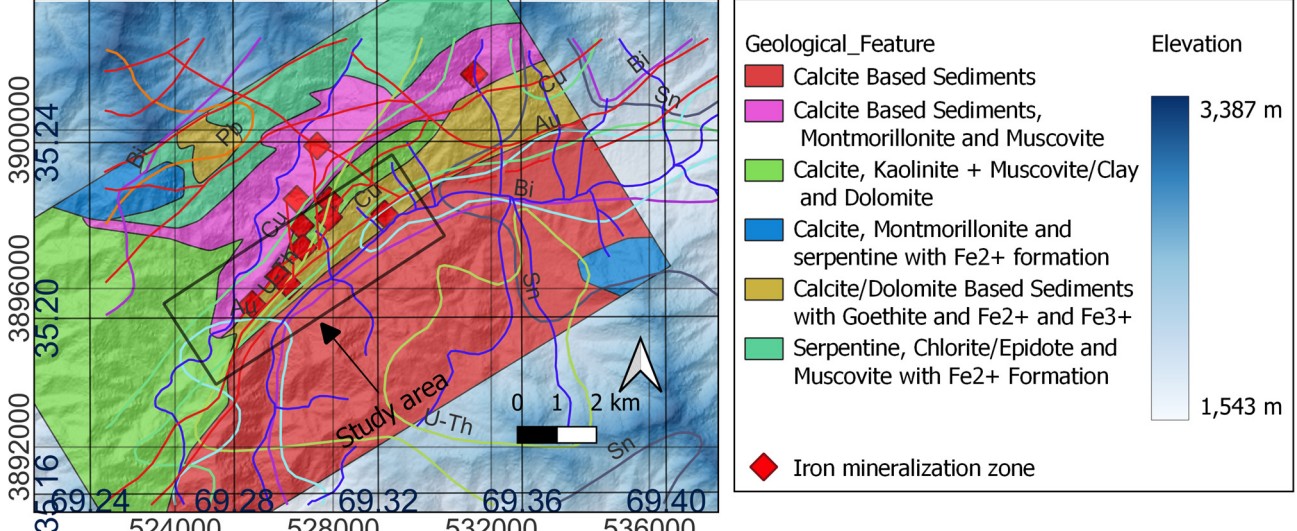

**Figure 2.** Geological map of the study area. The black-colored box on the map indicates the magnetic surveying conducted.

## 3. Magnetic Data

Magnetic data were measured in 2020 using a Cs vapor G-858 device with a 0.01 (nT) resolution. The survey was conducted along a northwest-southeast trend, with most data points separated at 25 m and lines over the polymetallic deposit at 1000 to 1500 m. International Geomagnetic Reference Fields and tidal variations were subtracted from the data. The resulting residual magnetic data were gridded using a 30-m cell size and a minimum-curvature algorithm, then combined into a single data set, Figure 3a.

To more accurately match magnetic anomalies with their causal sources, the remaining total magnetic field data were further processed using a reduction-to-pole (RTP) transformation [37]. RTP anomalies are closest to their origins when the total magnetization of the rock units is typically in line with the Earth's magnetic field as it is now. A 54.4° inclination and 3.5° declination were used while applying the RTP transformation. In the study area, the RTP map contrasts the highs and lows of magnetic anomalies in Figure 3b. Generally, the northeastern to southwestern trend in the study area is characterized by dominant positive magnetic anomaly highs with high amplitudes. Conversely, in the western side of this trend, there are lower amplitude magnetic fields, accompanied by lower and negative anomalies.

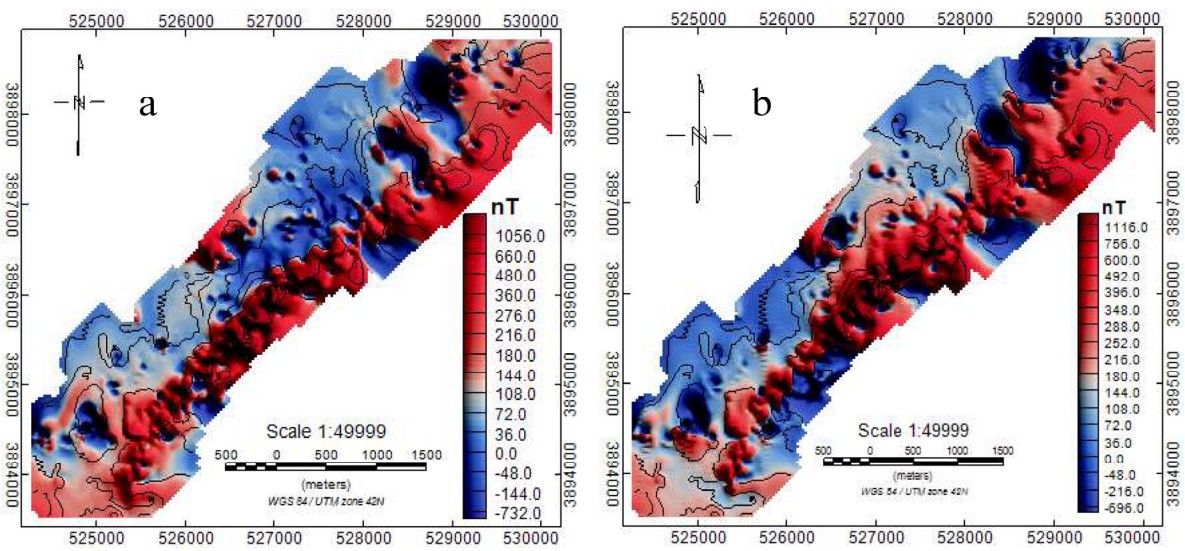

**Figure 3.** Total magnetic intensity map (**a**) and reduced to pole transformed map (**b**).

## 4. Multi-Scale Edge Detection

The multi-scale edge-detection approach was used to gain a general knowledge of the magnetic source geometry. The multi-scale edge detection involves extracting the analytical signal amplitude from a series of upward-continued data sets and identifying distinct summits. The findings could be employed to build a pseudo-3D image of sinking magnetic contacts. A complicated pattern may be discovered in the outcomes of multi-scale edge detection.

The procedure starts with an upward continuation [38,39]. In this stage, we elevate the magnetic data from its initial surface level to a greater height. This mathematical procedure helps reduce or attenuate the impact of shallow magnetic sources. Additionally, this transformation approach effectively decreases the interference from near-surface anomalies and isolates or accentuates the deeper magnetic sources. The next stage requires computing the analytical signal amplitude [40–42] on the magnetic data after the upward continuation transformation, which involves determining the analytical signal amplitude, which contains data on the strength or intensity of magnetic anomalies.

The analytical signal amplitude is useful for identifying underlying geological formations or resource potential and locations with large magnetic anomalies. By employing an upward continuation to reduce the influence of shallow sources and subsequently calculating the analytical signal amplitude, we enhance the ability to identify and interpret subsurface geological features and potential resource deposits from magnetic data. We calculate the analytical signal amplitude at each altitude once we have the upward continued magnetic data for various altitudes (e.g., 50, 100, 150, 200, and 300 units). Figure 4 illustrates the analytical signal amplitude applied on upward continued magnetic data. According to the regional geology map and analytical signal amplitude analysis, two prominent fault trends have been identified in the northeast to southwest direction. These fault trends are particularly distinct and noticeable in the analytical signal amplitude when it exceeds a threshold of upward of 300. In addition to the fault trends, multi-scale edge detection has also proven effective in enhancing the identification of other magnetic bodies located at the ends of the study area. The subsurface features, as identified through the analytical signal amplitude of the upward continuation data, are visually depicted in Figure 5.

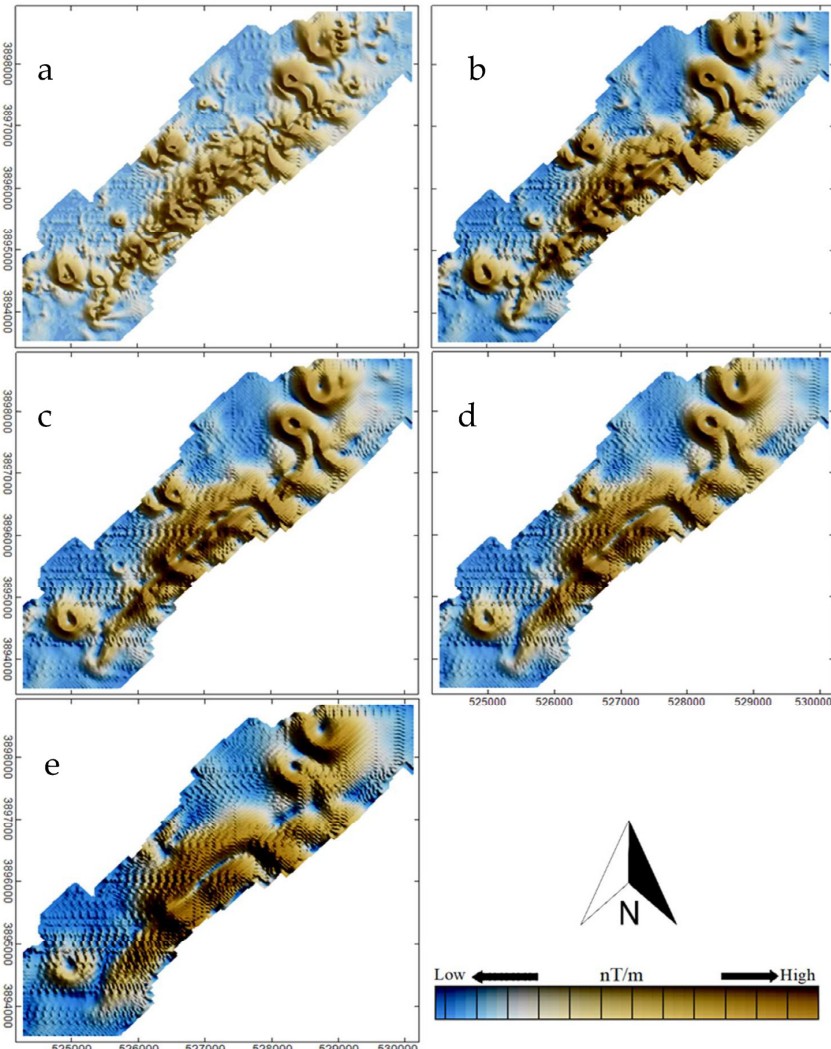

**Figure 4.** Analytical signal amplitude of the upward continued data in altitude (**a**) 50, (**b**) 100, (**c**) 150, (**d**) 200, and (**e**) 300. High-value trends identified through multi-scale edge detection can significantly enhance the delineation of magnetic body boundaries.

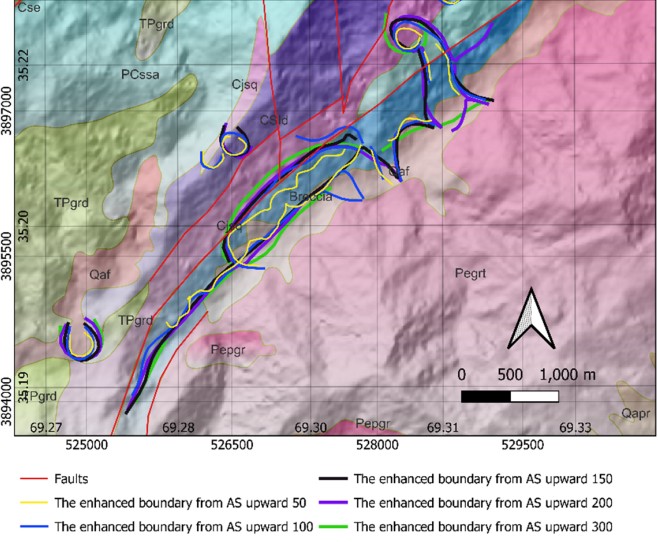

| | |
|---|---|
| —— Faults | —— The enhanced boundary from AS upward 150 |
| —— The enhanced boundary from AS upward 50 | —— The enhanced boundary from AS upward 200 |
| —— The enhanced boundary from AS upward 100 | —— The enhanced boundary from AS upward 300 |

**Figure 5.** The underground structures mapped by the analytical signal amplitude of upward continued data.

## 5. Magnetic Susceptibility Inversion

The initial inversion theory was built in Python and articulated by the work presented in [43] after being defined by the work presented in [44]. Smooth models are produced using the least-squares inversion. We applied the iteratively re-weighted least squares (IRLS), a specified technique for dealing with blocky/sparse models. Several authors, including the work presented in [45,46], have explained the well-defined inversion approach known as IRLS.

In SimPEG, a new module called the sparse norm inversion inverts the magnetic data to provide a susceptibility contrast model. As seen below, the forward simulation equation,

$$F(\boldsymbol{m}) = \boldsymbol{d}_{pred}, \tag{1}$$

where $\boldsymbol{d}_{pred}$ denotes the predicted information and $\boldsymbol{F}$ is the forward equation replicating the measurements using the following equation.

$$\boldsymbol{b}(r) = \frac{\mu_0}{4\pi} \int_V \nabla\nabla\frac{1}{r}\cdot\boldsymbol{M}dv, \tag{2}$$

where $r$ is the radial distance from any place to the magnetic source with magnetization per unit volume $\boldsymbol{M}$ (A/m), and $\boldsymbol{b}$ is the magnetic flux density in Tesla (T).

One of the most important components is the capability to model the geophysical problem and provide predicted data. After obtaining the predicted data, the $L_2$ norm is used to determine the data misfit.

$$\varphi_d(\boldsymbol{m}) = \frac{1}{2}|\boldsymbol{W}_d(F(\boldsymbol{m}) - \boldsymbol{d}_{obs}|_2^2, \tag{3}$$

where $\boldsymbol{W}_d$ is a diagonal matrix with the following elements:

$$w_{dii} = 1/\in_i, \tag{4}$$

where $\in_i$ is the estimated standard deviation of the *ith* datum.

The following norm expresses the definition of the model objective function.

$$\varphi_m(\boldsymbol{m}) = \alpha_s\|\boldsymbol{W}_s\boldsymbol{R}_s\left(\boldsymbol{m}-\boldsymbol{m}_{ref}\right)\|_2^2 + \alpha_x\|\boldsymbol{W}_x\boldsymbol{R}_x\boldsymbol{G}_x\left(\boldsymbol{m}-\boldsymbol{m}_{ref}\right)\|_2^2 + \alpha_y\|\boldsymbol{W}_y\boldsymbol{R}_y\boldsymbol{G}_y\left(\boldsymbol{m}-\boldsymbol{m}_{ref}\right)\|_2^2$$
$$+\alpha_z\|\boldsymbol{W}_z\boldsymbol{R}_z\boldsymbol{G}_z\left(\boldsymbol{m}-\boldsymbol{m}_{ref},\|\right)_2^2, \tag{5}$$

where $s$, $x$, $y$, and $z$ are the variables that determine the relative weights of each of the four terms, $R_s$, $R_x$, $R_y$, and $R_z$ define the norms, and $G_x$, $G_y$, and $G_z$ define the gradients in each direction. Considering the Equation (5), the regular function in matrix form is:

$$\varphi_m(\boldsymbol{m}) = \sum_{r=s,x,y,z} \alpha_s\|\boldsymbol{W}_r\boldsymbol{R}_r\boldsymbol{G}_r\boldsymbol{m}\|_2^2. \tag{6}$$

The optimization is defined as follows, given the model objective functions and the data misfit:

$$minimize\ m,\ \ \varphi(\boldsymbol{m}) = \varphi_d(\boldsymbol{m}) + \beta\varphi_m(\boldsymbol{m})$$
$$such\ that\ \ \ \varphi_d \leq \varphi_d^*\ \ \ \ \ \ \ m_i^L \leq m_i \leq m_i^H, \tag{7}$$

where the term $\beta$ "trade-off parameter" refers to the regularization term.

## 6. Magnetic Vector Inversion (in Cartesian and Spherical Formula)

Following the work presented in [47], based on the observed total magnetic field intensity (TMI) data, we provide a process for extracting information about the subsurface

magnetization vector model. The total magnetic field per unit volume may be divided into its induced and remanent components in a way that:

$$\boldsymbol{M} = \kappa(\boldsymbol{h}_0 + \boldsymbol{h}_s) + \boldsymbol{M}_r, \tag{8}$$

where the physical property defining a rock's capacity to get magnetized by an applied field is called magnetic susceptibility $\kappa$ (SI). The geomagnetic field of the earth, or $\boldsymbol{h}_0$, and secondary fields, or $\boldsymbol{h}_s$, related to local magnetic anomalies, make up this inducing field. By ignoring the remanent and self-demagnetization effects and assuming that the magnetic response is only generated along the Earth's field ($\boldsymbol{M}_r = \boldsymbol{h}_s = 0$), the magnetization Equation (8) can be defined as:

$$\boldsymbol{M} = \kappa(\boldsymbol{h}_0). \tag{9}$$

So, rewriting the Equation (1), linear system relating $N$ data, $\boldsymbol{d}_{pred}$, to $M$ discrete model cells of magnetic susceptibility $\kappa$ should be as:

$$\boldsymbol{d}_{pred} = \boldsymbol{F}\kappa. \tag{10}$$

Ref. [48] describe an effective susceptibility parameter that scales the strength of magnetism in orthogonal directions without making any assumptions about orientation to recover the magnetization vector. So, the Equation (8) is:

$$\kappa_e = \frac{\boldsymbol{M}}{\|\boldsymbol{h}_0\|}. \tag{11}$$

The augmented system may be generated by rewriting the discrete system in Equation (10) in terms of the three orthogonal components of magnetization ($u$, $v$, *and* $w$), two components perpendicular to the Earth's field and one component parallel to it in the Cartesian formulation.

$$\boldsymbol{d}_{pred} = [\boldsymbol{F}_u \boldsymbol{F}_v \boldsymbol{F}_w] \begin{bmatrix} \kappa_u \\ \kappa_v \\ \kappa_w \end{bmatrix}, \tag{12}$$

where the forward operators for the components of magnetization are $\boldsymbol{F}_u$, $\boldsymbol{F}_v$, and $\boldsymbol{F}_w$. In contrast to the susceptibility assumption, (47) works with a linear system with three times as many unknown parameters $\kappa_e^{3\times\mathbb{R}}$. In Equation (8), the regularization function is transformed to:

$$\varphi_m(\boldsymbol{m}) = \sum_{c=u,v,w} \sum_{r=s,x,y,z} \alpha_s \|\boldsymbol{W}_{c_r} \boldsymbol{R}_{c_r} \boldsymbol{G}_{c_r} \boldsymbol{P}_c \kappa_e\|_2^2, \tag{13}$$

where the projection matrices $\boldsymbol{P}_c$ choose up certain elements of the vector model $\kappa_e$. We have 12 terms for regularization. For each Cartesian component, norm measures can be used separately.

The relationship that dictates how the Cartesian system is changed into a spherical system is:

$$u = \rho\cos(\theta)\cos(\phi), \ v = \rho\cos(\theta)\sin(\phi), \ w = \rho\sin(\theta). \tag{14}$$

Here, the amplitude ($\rho$) and two angles ($\theta$, $\phi$) parameters define the magnetization vector. With this parameterization selection, in the spherical formula, the regularization function changes to:

$$\varphi_m(\boldsymbol{m}) = \sum_{c=\rho,\theta,\phi} \sum_{r=s,x,y,z} \alpha_s \|\boldsymbol{W}_{c_r} \boldsymbol{R}_{c_r} \boldsymbol{G}_{c_r} \boldsymbol{P}_c \boldsymbol{m}\|_2^2. \tag{15}$$

## 7. Model Setup and Inversion

As a starting point, our initial assumption is that no remnant magnetization is present in the geological materials under investigation. Instead, we focus solely on induced magnetization to recover the susceptibility model. In the process of conducting magnetic

susceptibility inversion, a crucial step involves the discretization of our study area into a well-structured grid. We employ a tensor mesh approach to achieve this, resulting in a three-dimensional grid with dimensions of $80 \times 80 \times 80$ m. This finely detailed mesh is instrumental in our efforts to invert the magnetic data effectively, allowing us to analyze and interpret magnetic susceptibility properties across the study area with precision and accuracy. Therefore, we used the reduced-to-pole magnetic data in Figure 3b as a fundamental dataset. We employed a target misfit of the number of data points with a *chi factor* $= 1$, a standard error ($\in$) = 95 nT plus 2% of the data range. After 38 iterations, the magnetic susceptibility inversion converged to this floor, achieving a stable model norm $\varphi_m$. The residual from the inversion with norm [0,1,1,1] with unitless normalization is shown in Figure 6a–c as [(observed data-predicted data)/standard deviation].

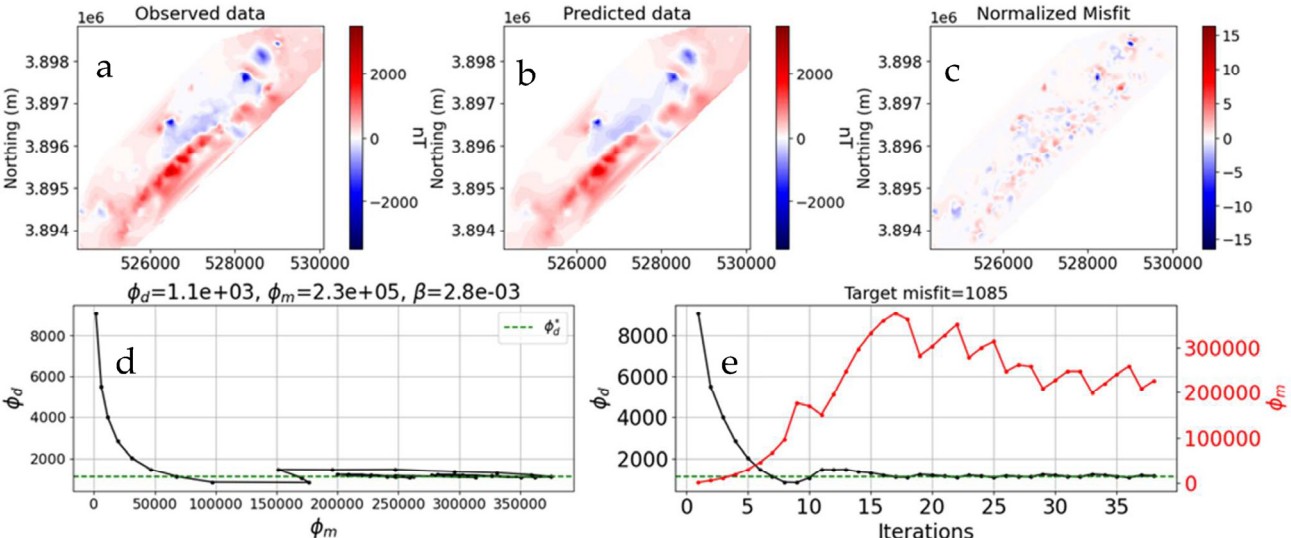

**Figure 6.** (**a**) Observed data (reduced to pole), (**b**) Predicted data calculated from the model norm [0,1,1,1] in magnetic susceptibility inversion, (**c**) Normalized residual spatial distribution. (**d**) Inversion convergence curves for the model norm [0,1,1,1] with model size ($\varphi_m$) and misfit ($\varphi_d$), and (**e**) data misfit behavior in black and model norm in red with iterations. The green-dashed line shows the target misfit.

In the subsequent phase of our study, we expanded our analysis to incorporate the presence of remnant magnetization. To address this, we employed a specialized technique known as magnetization vector inversion to invert total magnetic intensity data, as seen in Figure 3a. This approach allowed us to account for induced and remnant magnetization, providing a more comprehensive and accurate assessment of the magnetic properties within our study area. Hence, we adopted a comprehensive approach by considering Cartesian and spherical coordinate frameworks. For the magnetization vector model, we used an octree mesh to discretize the study area, in contrast to our method for magnetic susceptibility inversion. A complex procedure available through SimPEG is employed to create an octree mesh. The predicted data at the location is then obtained by calculating the magnetic effect of each cell at the designated measurement point and adding up all of the effects of each cell. With dimensions of $120 \times 120 \times 120$, this meshing approach was used to ensure a more accurate and focused study of the magnetization vectors. These dimensions were chosen to match the magnetic data's spatial distribution closely. We set the regularization parameters for the Cartesian and spherical coordinate systems as [0000,0000,0000] magnetization vector inversion. We set a target misfit by aligning the number of data points with a *chi factor* $= 1$, and a standard error ($\in$) of 50 nT, plus 2% of the data range. Following 12 iterations, the magnetization vector inversion, utilizing the Cartesian formula, reached convergence at this threshold. As a result, it attained a stable model norm $\varphi_m$, as illustrated in Figure 7c.

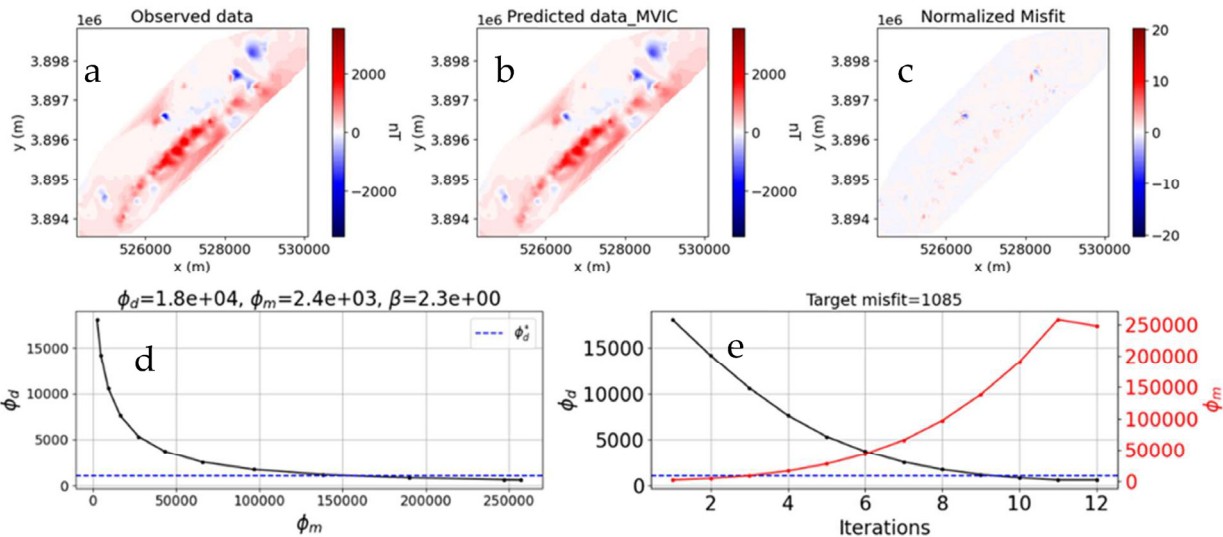

**Figure 7.** (**a**) Observed data (total magnetic intensity), (**b**) Predicted data calculated from the model norm [0000,0000,0000] in magnetization vector inversion using the Cartesian framework, (**c**) Normalized residual spatial distribution. (**d**) Inversion convergence curves for the model norm [0000,0000,0000] with model size ($\varphi_m$) and misfit ($\varphi_d$), and (**e**) data misfit behavior in black and model norm in red with iterations. The blue dashed line shows the target misfit.

We established the magnetization vector model from the Cartesian formula as our reference model for the magnetization vector inversion conducted in spherical coordinates. Figure 8 illustrates the predicted data, which has been calculated through the process of magnetization vector inversion within the spherical framework. The regularization parameters used in the spherical formula are the same as those applied in the Cartesian formula.

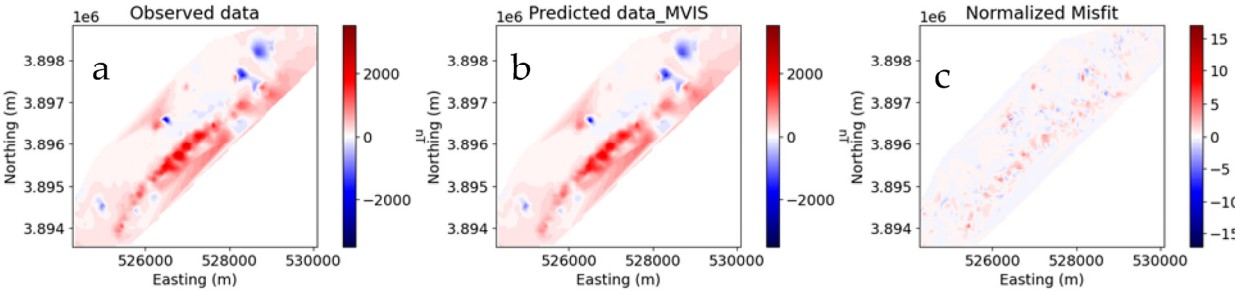

**Figure 8.** (**a**) Observed data (total magnetic intensity), (**b**) Predicted data calculated from the model norm [0000,0000,0000] in magnetization vector inversion using Spherical framework, (**c**) Normalized residual spatial distribution.

## 8. Magnetic Susceptibility Model

The slices are presented in horizontal and vertical orientations, showcasing the subsurface at different depths and horizontal directions (Figure 9). These orientations allow us to examine the horizontal and vertical variations in magnetic susceptibility. The figure employs a color scale to represent magnetic susceptibility values. Warmer colors like red indicate higher magnetic susceptibility, while cooler colors like blue represent lower values. This color scheme facilitates the identification of regions with varying magnetic properties. Throughout the slices, distinct magnetic susceptibility anomalies are evident. Warmer colors characterize positive anomalies and signify areas with elevated magnetic susceptibility. Conversely, negative anomalies in cooler colors suggest regions with reduced magnetic susceptibility compared to the surrounding material.

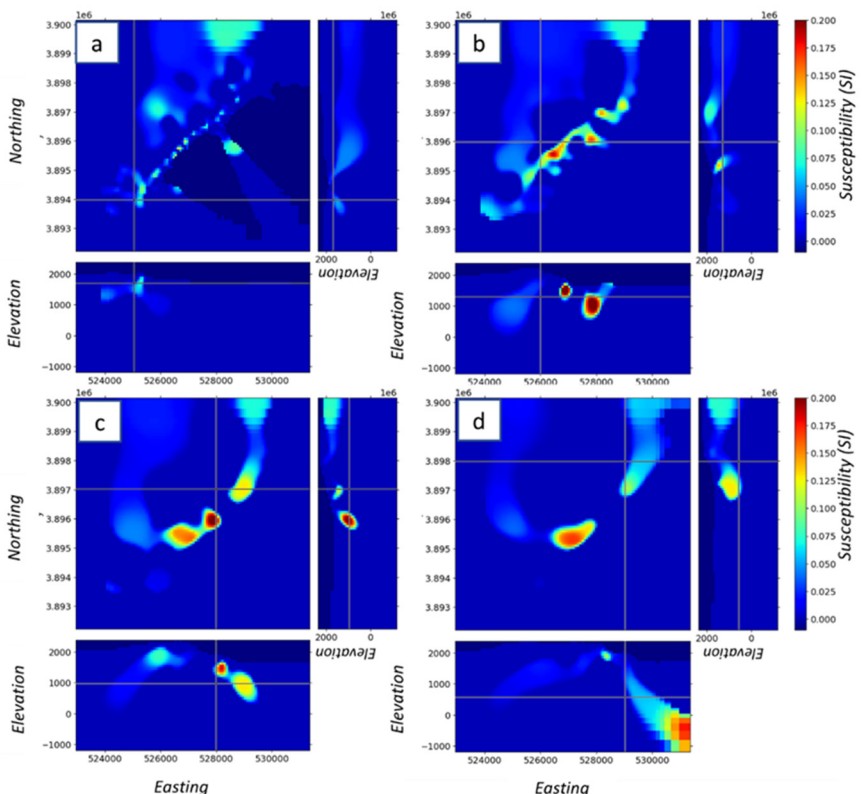

**Figure 9.** The horizontal and vertical slices of the recovered susceptibility model. (**a**) (x = 525,000, y = 3,894,000, and z = 1700), (**b**) (x = 526,500, y = 3,896,000, z = 1300), (**c**) (x = 528,200, y = 3,897,000, and z = 1000), and (**d**) (x = 529,500, y = 3,898,000, and z = 600). Gray lines along the x, y, and z axes in various panels represent the vertical and horizontal cross sections.

Figure 9a presents a detailed representation of the magnetic susceptibility model, offering insights into the subsurface properties at the specific coordinates of x = 525,000, y = 3,894,000, and z = 1700 m below the Earth's surface. This figure combines horizontal and vertical slices, providing a comprehensive view of the magnetic susceptibility distribution at this precise location. The figure includes a horizontal slice that captures the variation in magnetic susceptibility as it moves laterally within the subsurface at the specified depth (z = 1700). This horizontal slice is typically parallel to the Earth's surface at the given depth. Vertical slices are presented, providing a cross-sectional view of the magnetic susceptibility distribution in the subsurface. These vertical slices allow us to examine the lateral variation in magnetic properties at the specified sections and locations. At this depth, a scattered magnetic body associated with iron-bearing polymetallic mineral distribution is observed, forming a continued trend running from northeast to southwest.

In Figure 9b, the magnetic susceptibility model continues to provide insights into subsurface properties, focusing on the coordinates (x = 526,500, y = 3896,000, z = 1300) meters below the Earth's surface. Similar to the previous panel, this figure combines horizontal and vertical slices, offering a comprehensive view of the magnetic susceptibility distribution at this precise location. This figure includes horizontal slices depicting how magnetic susceptibility varies laterally within the subsurface at the specified coordinates. The horizontal slice is situated at a depth of 1300 m (z = 1300). Vertical slices are presented, giving us a cross-sectional perspective of the magnetic susceptibility distribution within the subsurface. As we delve to a depth of 1300 m units, we encounter magnetic bodies of increased magnitude, offering significant information regarding lateral variations along the continuous trend that extends from northeast to southwest.

Focusing on the coordinates (x = 528,200, y = 3,897,000, z = 1000) meters below the Earth's surface, the magnetic susceptibility model sheds light on subsurface characteristics in Figure 9c. Like the earlier panels, this image contains horizontal and vertical slices to

provide a thorough perspective of the magnetic susceptibility distribution at this site. The horizontal slices in this picture section show the lateral variations in magnetic susceptibility inside the subsurface at the given coordinates. A depth of 1000 m (z = 1000) separates this slice from the surface. The magnetic susceptibility distribution inside the subsurface is also shown in vertical slices, giving a cross-sectional perspective. The lateral changes in the magnetic properties at the particular depth and position can be examined using these slices. At a depth of 1000 m, our findings revealed the presence of exclusively massive magnetic bodies in central part of the study area, distinctly linked to high magnetic susceptibility.

By providing a three-dimensional image of the ground beneath the specific coordinates of (x = 529,500, y = 3,898,000, z = 600), Figure 9d deepens the investigation into the distribution of magnetic susceptibility. The full perspective of the magnetic susceptibility distribution at a single location inside the subsurface shown in Figure 9d completes the larger geophysical investigation. It helps create a full understanding of the local subsurface geology. It is a reasonable interpretation to consider that the highly susceptible areas, as indicated by magnetic susceptibility measurements, may potentially contain iron-prospective geological formations. Elevated magnetic susceptibility often correlates with iron-rich minerals or deposits, making these areas promising targets for further exploration and assessment of iron resources. Figure 10 provides a three-dimensional perspective, allowing us to explore the spatial distribution of values within the dataset at or below 0.06 SI. That can reveal geological and geophysical features' shape, extent, and spatial relationships. To obtain a more comprehensive understanding of the subsurface structure and geological formations affected by remnant magnetization, we have successfully recovered the magnetization vector model.

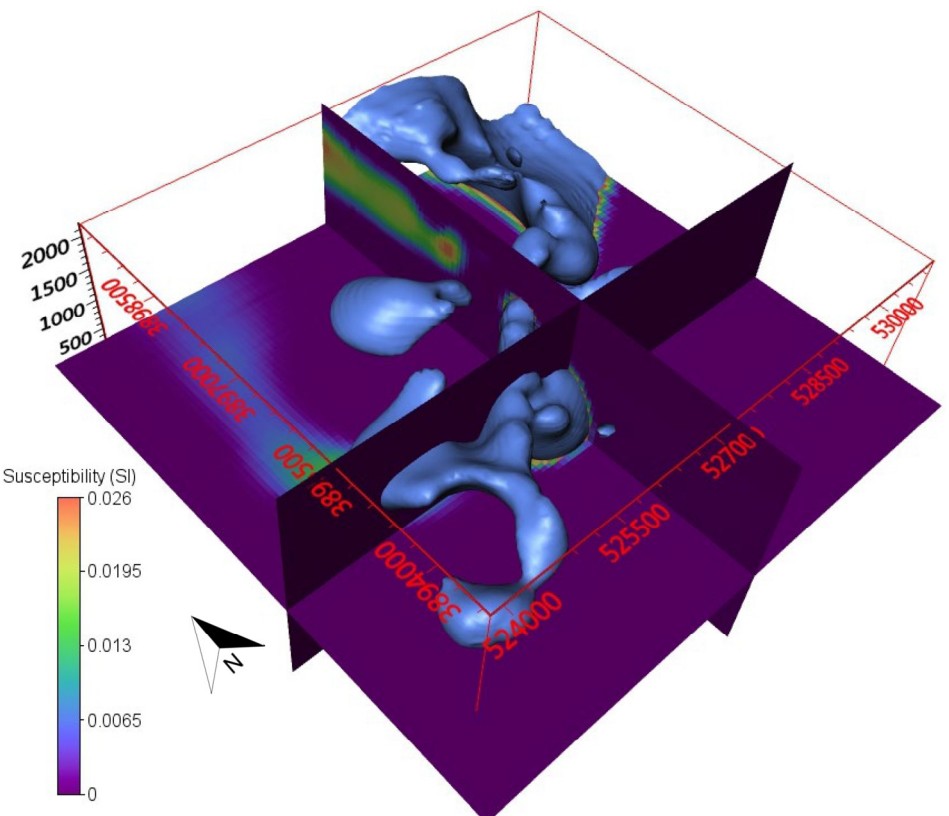

**Figure 10.** A 3D visualization is presented, showing an isosurface within a geophysical dataset. This isosurface represents values equal to or below 0.026 SI units.

## 9. Magnetization Vector Model

The magnetization vector model, elegantly presented in both horizontal and vertical slices in Figures 11–13, offers a multi-dimensional perspective into the distribution and

orientation of magnetization vectors. Within each slice, the amplitude and strength of magnetization are eloquently expressed through the length of arrows. Longer arrows indicate areas of heightened magnetization, highlighting regions of particular interest. These zones may correspond to concentrations of magnetic minerals, ore bodies, or geological structures with distinct magnetic properties. The color scale was also designed with a particular objective: to recognize and differentiate higher magnetic objects depending on the strength of their magnetization. The warm red color serves as a sign and a marker for the locations where magnetization vectors and effective susceptibility are strongest. These areas can be identified by the Earth's magnetic field being strongly influenced by magnetic minerals or geological features.

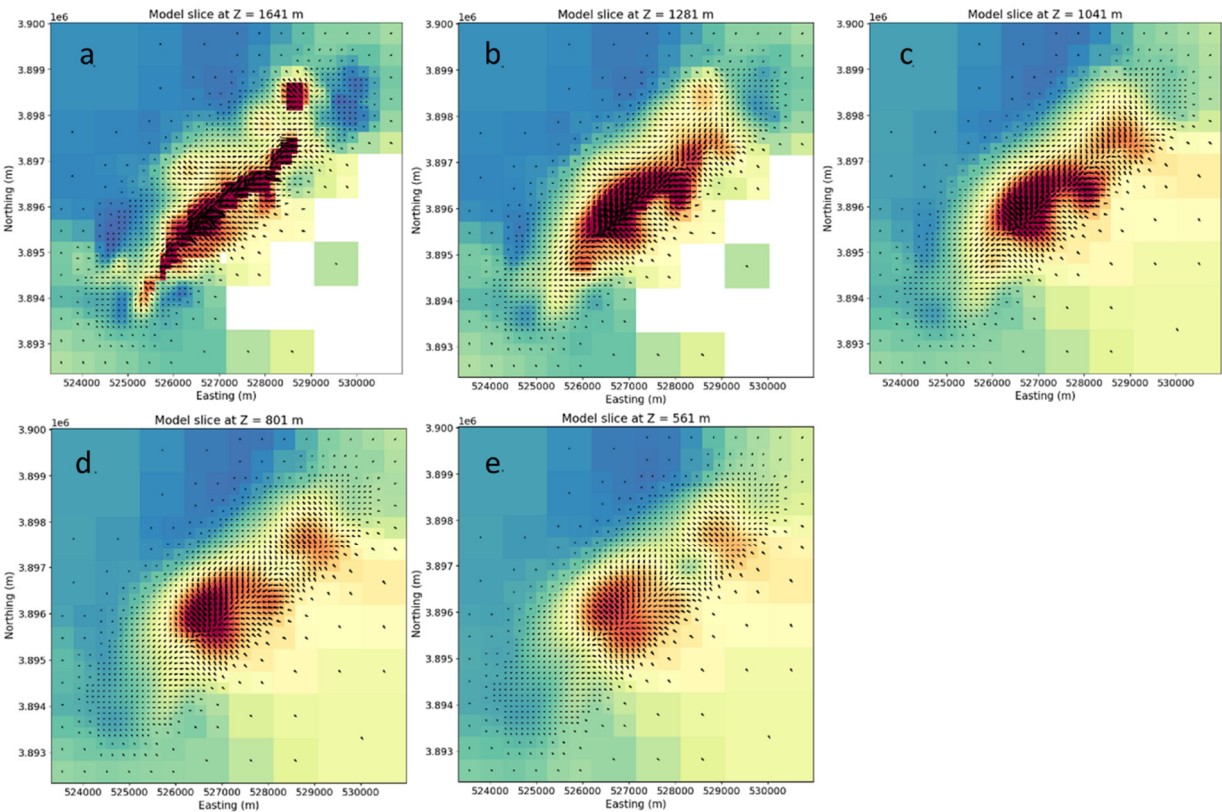

**Figure 11.** The horizontal slices at distinct depths within the subsurface of the study area. The slices are positioned at (**a**) z = 1641 m, (**b**) z = 1281 m, (**c**) z = 1041 m, (**d**) z = 801 m, and (**e**) z = 561 m.

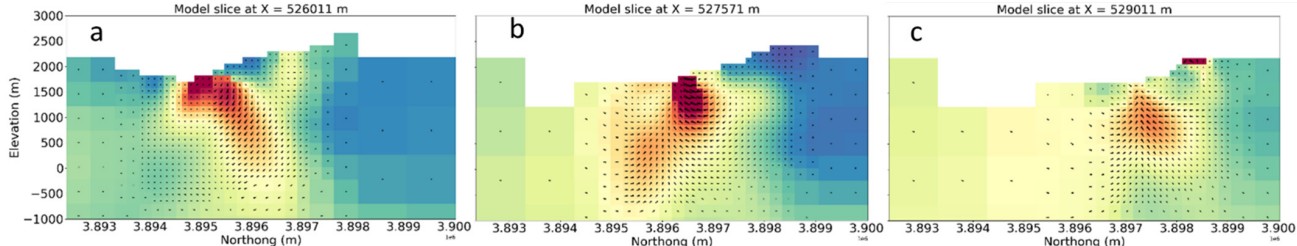

**Figure 12.** Illustration of vertical slices along the south-north direction at (**a**) x = 526,011 m, (**b**) x = 527,571 m, and (**c**) x = 529,011 m.

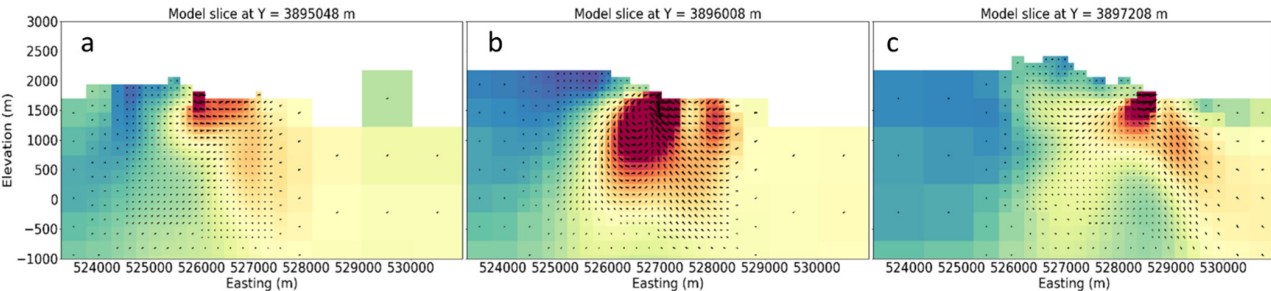

**Figure 13.** Three vertical slices along the east-west direction, at (**a**) y = 3,895,048 m, (**b**) y = 3,896,008 m, and (**c**) y = 3,897,208 m.

Figure 11 shows several horizontal cross-sections of the subsurface that resemble the Earth's layers being sliced to the ground. The magnetization vector model assumes a planar shape in these slices, displaying the distribution and direction of magnetization vectors at certain depths. With the help of this horizontal image, we can see how the magnetization vectors change with depth and get important knowledge about the underlying geological features. The vertical slices in Figures 12 and 13 complement our horizontal view by offering a side-on view similar to a geological cross-section. Here, we gain a new perspective to observe the magnetization vectors' distribution and lateral orientation within the subsurface. Understanding the three-dimensional intricacy of geological formations is very helpful with this viewpoint. Our study via these horizontal and vertical slices of our magnetization vector model takes us deep within the area's geological tapestry. Each slice offers a special view into a distinct part of the subsurface, enhancing our knowledge and adding to the overall story of our investigation. Given the absence of prior geological information, we infer that the areas exhibiting high values in magnetization and significant vector data can be interpreted as probable zones for Fe-polymetallic deposits.

## 10. Conclusions

In our research, we conducted a magnetic survey investigation to map the perspectivity of Fe-polymetallic deposits in Panjshir, Afghanistan. This magnetic survey is an essential step in assessing the potential presence of valuable iron and polymetallic minerals, contributing to our understanding of the mineral resources in the area. We have successfully identified and delineated the boundaries of magnetic bodies and major fault systems within the study area by employing multi-scale edge detection techniques. This approach enhances our ability to precisely locate and characterize these geological features, which is crucial for mineral exploration and geological research.

The utilization of magnetic susceptibility inversion has proven instrumental in our research, facilitating the mapping of the spatial distribution and extent of magnetic bodies closely associated with iron-bearing geological formations. We have employed the magnetization vector inversion method to acknowledge the challenges of remnant magnetization in mapping deeper geological structures. This approach enables us to effectively map deeper magnetic bodies even in remnant magnetization, overcoming the limitations typically associated with conventional magnetic data inversion. In the models generated through magnetic susceptibility inversion and magnetization vector inversion, areas characterized by high susceptibility values and high magnetization magnitudes indicate the presence of iron-bearing polymetallic formations in the subsurface.

**Author Contributions:** M.H.R., A.Q.A., T.P., E.R. and K.M.S. contributed substantially to various aspects of the research, including conceptualization, methodology, software development, data collection, writing, original draft preparation, visualization, and conducting the research. Supervision, conceptualization, methodology development, investigation, reviewing, and editing were all undertaken by H.M. All authors have read and agreed to the published version of the manuscript.

**Funding:** This research received no external funding.

**Data Availability Statement:** Upon a reasonable request, the corresponding author will make the data sets and inversion code used in this research available.

**Acknowledgments:** The first author would like to express their sincere appreciation to the JICA (Japan International Cooperation Agency) for their generous support in the form of a scholarship. This support has been invaluable in facilitating and advancing our research efforts.

**Conflicts of Interest:** The authors declare no conflict of interest.

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
