# Peer review of "3D Geophysical Modeling Based on Multi-Scale Edge Detection, Magnetic Susceptibility Inversion, and Magnetization Vector Inversion in Panjshir, Afghanistan to Detect Probabilistic Fe-Polymetallic Bearing Zone"

_geosciences, doi:10.3390/geosciences13120376_

Round 1
Reviewer 1 Report
Comments and Suggestions for Authors
Please see the attached document.

Author Response
Subject: Response to Reviewer's Comments
Dear Sir/Madam,
I trust this message finds you well. Please find attached the file containing my detailed response to your insightful comments. I appreciate your time and thoughtful feedback, and I have addressed each comment in the attached document.
Thank you for your valuable input, and I look forward to any further guidance or suggestions you may have.
Best regards,
Mohammad Hakim Rezayee

Reviewer 2 Report
Comments and Suggestions for Authors
Title: please correct: Fe-Polymetllic
I would recommend to adjust the title to the article content: multi-scale edge detection, magnetic susceptibility inversion and magnetization vector inversion
Magnetic susceptibility is a measure of how easily a material can be magnetized. Different geological materials have distinct magnetic susceptibilities, and variations in these properties can be indicative of changes in subsurface composition. Inversion is the mathematical process used to estimate the distribution of magnetic susceptibility within the subsurface.
Magnetic vector inversion involves estimating both the magnitude and orientation (vector components) of the magnetization within the subsurface.
The integrated approach of magnetic susceptibility inversion and magnetic vector inversion provides a more detailed and comprehensive understanding of subsurface magnetic properties, making it valuable for mineral exploration, environmental studies, and geological mapping.
The resulting 3-D model provides a visual representation of the spatial variations in magnetic susceptibility and magnetization vector components within the subsurface.
Lines 17-18: Due to high magnetic property, geomagnetic data were utilized to map the probabilistic prospectivity of Fe-polymetallic deposits in the study area. – Please rephrase, e.g.: Considering their high magnetic properties, […]
Lines 39-40: Please rephrase: At the moment, the top three and fourth most utilized metals in the world are iron, copper, and zinc, respectively.
Please rephrase: Figure 1. The Middle Afghanistan Geosuture (Chmyriov et al., 1982), which trends westward through Bamyan and separates the platform sedimentary rocks found in northern Afghanistan from southern structural blocks of various origins, plays a significant role in the identification of the geology in the Panjshir Valley and Western Hindu Kush region. (Lines 94-97)
Figure 1. (a) – needs to be enlarged
Figure 1. (b) – no clearly visible geology, no clearly visible faults (marked with red lines in legend), no clearly visible iron-bearing zone (marked with red rhomb in legend) – needs to be redrawn, with less transparency of the geology layer. Yellow box needs to be drawn in thicker line, for visibility – no need for the red arrow and study area writing, as it is stated in the figure caption.
Figure 1 – Legend is not readable
Lines 132-142: We have emphasized the iron-bearing zone using remote sensing techniques, particularly in satellite image processing. This approach involves analyzing and interpreting satellite imagery to gain valuable insights into the characteristics and distribution of iron-rich areas within this zone. A cloud-free level 1T ASTER image was downloaded from the U. S. Geological Survey Earth Resources Observation and Science Center (USGS EROS) (https://earthexplorer.usgs.gov). A band combination of 4,6,8 is used to visualize the geo-logical formation, and the sediment boundaries are divided based on a color variation of the sediments. Using remote sensing techniques, we have identified and delineated litho-logical boundaries within the area of interest (Kaufman and Remer, 1994; Vincent,2004; Abrams, 1993). Subsequently, we have refined and constrained these boundaries by utilizing a hyperspectral map from (Trude et al., 2013), allowing us to gain a more detailed understanding of the mineral composition in the study area Figure 2. – no clear connection of the technique explained in this paragraph to Figure 2 - Geological map of the study area. Paragraph needs to be rephrased and better explained the author’s contribution to the figure.
Figure 2. – no clearly visible faults (marked with red lines in legend), no clearly visible iron-bearing zone (marked with red rhomb in legend). Yellow box needs to be drawn in thicker line, for visibility – no need for the red arrow and study area writing, as it is stated in the figure caption. Legend is not readable
Lines 164-166: In general, high-amplitude magnetic anomaly highs positive anomaly dominated in the northeastern to southwestern trend, whereas moderate amplitude magnetic highs the lower and negative anomaly occurred further in central and western parts of the study area. – needs rephrasing for the clarity of ideas and better understanding
Figure 10. – no clear content – please consider enlarging and presenting figures 10-a and 10-b subsequently. I also recommend adding the cross-sections’ location on a detail map, for positioning.
Figures 11-13: text not readable
Figures 12-13: I recommend adding the cross-sections’ location on a detail map, for positioning.
Comments on the Quality of English LanguageThe text is difficult to read and further understand the flow of ideas and argumentation. Extended rephrasing for clarity is needed.
Author Response

(The authors gave the same response as above.)

Round 2
Reviewer 2 Report
Comments and Suggestions for Authors
Figure 1 - Legend appears first, on different page than map features.
Author Response
Dear Reviewer,
Thank you sincerely for dedicating time to review our manuscript. We appreciate your insightful comments, and we have carefully considered each suggestion. Below, we present detailed responses along with corresponding revisions highlighted or tracked in the re-submitted files.
Summary of Changes:
Comments: Figure 1 - Legend appears first, on different page than map features.
Response: In response to your suggestion, we have adjusted the position of the legend and the geological map to enhance clarity and readability. Currently, the map appears first, followed by the legend in the revised version. Despite this adjustment, we acknowledge that the legend and geological map are still on different pages. There are detailed explanations on the map and legend bar. If we adjust to a smaller size, there is a risk of decreased resolution and clarity for both the map and its legend.
We believe these revisions have significantly strengthened the manuscript in accordance with your valuable feedback.
Thank you for your continued consideration.
Best regards,
Mohammad Hakim Rezayee